# A generic approach to study the kinetics of liquid–liquid phase separation under near-native conditions

Joris Van Lindt [1,2,7], Anna Bratek-Skicki[1,2,7✉], Phuong N. Nguyen [1,2,3], Donya Pakravan[4,5], Luis F. Durán-Armenta [1,2], Agnes Tantos[6], Rita Pancsa[6], Ludo Van Den Bosch [4,5], Dominique Maes [2] & Peter Tompa [1,2,6✉]

Understanding the kinetics, thermodynamics, and molecular mechanisms of liquid–liquid phase separation (LLPS) is of paramount importance in cell biology, requiring reproducible methods for studying often severely aggregation-prone proteins. Frequently applied approaches for inducing LLPS, such as dilution of the protein from an urea-containing solution or cleavage of its fused solubility tag, often lead to very different kinetic behaviors. Here we demonstrate that at carefully selected pH values proteins such as the low-complexity domain of hnRNPA2, TDP-43, and NUP98, or the stress protein ERD14, can be kept in solution and their LLPS can then be induced by a jump to native pH. This approach represents a generic method for studying the full kinetic trajectory of LLPS under near native conditions that can be easily controlled, providing a platform for the characterization of physiologically relevant phase-separation behavior of diverse proteins.

---

[1] VIB-VUB Center for Structural Biology, VIB, Brussels, Belgium. [2] Structural Biology Brussels, Vrije Universiteit Brussel, Brussels, Belgium. [3] Department of Biology, College of Natural Sciences, Cantho University, Can Tho, Vietnam. [4] VIB, Center for Brain & Disease Research, Laboratory of Neurobiology, Leuven, Belgium. [5] KU Leuven, Department of Neurosciences, Experimental Neurology and Leuven Brain Institute (LBI), Leuven, Belgium. [6] Institute of Enzymology, Research Centre for Natural Sciences, Budapest, Hungary. [7] These authors contributed equally: Joris Van Lindt, Anna Bratek-Skicki. ✉email: anna.bratek-skicki@vub.be; peter.tompa@vub.be

Compartmentalization of eukaryotic cells via organelles is important for the spatiotemporal separation of their biochemical reactions. Many organelles in the cell are surrounded by membranes that physically separate them from the bulk of the cytoplasm, whereas newly discovered membraneless organelles (MLOs) lack such a physical barrier, yet they are involved in basic cellular activities. A surge of recent studies have suggested that the primary mechanism for the creation of these supramolecular assemblies is liquid–liquid phase separation (LLPS)[1–3]. For example, stress granules rapidly form by LLPS upon cellular stress and play important roles in cell survival. Due to their central function in cell physiology, a loss of control over their formation and clearance is also linked to neurodegenerative diseases, such as amyotrophic lateral sclerosis (ALS)[4,5]. In vitro, stress-granule proteins, such as TAR DNA-binding protein 43 (TDP-43), fused in sarcoma (FUS), and heterogeneous nuclear ribonucleoprotein A2/B1 (hnRNPA2/B1) form liquid droplets, then undergo gelation and may be converted into aggregated fibrils[5,6]. Similar phase transitions were observed for the nuclear pore complex (NPC) protein NUP98, which plays an important role in the bidirectional transport of macromolecules across the NPC[7]. Interestingly, stress proteins such as Early Responsive to Dehydration 14 (ERD14), can also undergo functional LLPS[8]. Therefore, understanding the biophysical principles that govern LLPS is a central goal in diverse areas of current cell biology research.

Studies of LLPS are dominated by visualizing mature liquid droplets at a stage considered to correspond to equilibrium. Strictly speaking, however, phase-separated droplets are never in a true thermodynamic equilibrium, i.e., characterizing the kinetics of their nucleation and subsequent evolution are of paramount importance for understanding their mechanisms of formation and regulation. The proteins involved in these processes, however, are often severely aggregation-prone and are usually prepared and kept in solution either with a fused solubility tag (MBP, GFP, or GST[9,10], or under denaturing (6–8 M urea)[11] or otherwise non-physiological (very high salt, detergents) conditions[12]. Although often disregarded, these additional factors may strongly compromise protein behavior. For example, phase separation of the prion-like domain (PLD) of the physiological yeast prion, Sup35, has been studied under two different initial states, starting from highly denaturing conditions (8 M guanidine hydrochloride) or from a stock of very high concentration of salt (1 M KCl)[12]. When LLPS experiments started from denaturing conditions, Sup35 PLD aggregated into amyloid-like fibers[13], whereas when it was diluted from high salt, it phase-separated into liquid droplets which later turned into gel-like condensates[14]. This and other examples highlight that the path of LLPS is very sensitive to experimental conditions, and may take different kinetic trajectories proceeding from the solution state to liquid droplets and then to gels and amorphous or amyloid aggregates.

To develop reproducible, physiologically relevant, and specific mechanistic models, we propose here a generic and easily adaptable method to study the kinetic path of LLPS, in which proteins are kept in solution at a carefully selected extreme pH and their phase separation is induced by a jump to physiological pH. With this approach, a change in conditions favoring LLPS occurs instantaneously and entails no dilution effect, the benefits of which are demonstrated via comparing it to other approaches, such as inducing LLPS by dilution of the protein from highly denaturing conditions (8 M urea) or cleavage of its fused solubility tag (maltose binding protein, MBP). We show that the actual conditions have a profound effect on the process of LLPS, leading to very different mechanistic conclusions.

## Results

### Phase separation of the low-complexity domain of hnRNPA2.

First, we compare the LLPS of the low-complexity domain (LCD) of hnRNPA2 upon a pH jump with those initiated by diluting the protein from a solution of 8 M urea and by cleaving off its fused MBP solubility tag[11]. For the pH jump, we can keep the protein in solution at pH 11.0, and then change its pH to 7.5 by a small amount of concentrated buffer (cf. Methods), in a way similar to that suggested for inducing LLPS[15] or aggregation[16] of highly aggregation-prone proteins. Small droplets quickly formed, then grew over time, and even turned into aggregates in 2 h (Fig. 1a). LLPS could be reversed by increasing the pH back to 11.0 (Fig. S1). The reaction was slowed down by salt: in the presence of 150 mM NaCl, only small droplets formed (Fig. 1a). To demonstrate that different conditions, even under the same final protein concentration, lead to very different kinetic schemes, we next monitored the progress of the LLPS of hnRNPA2 LCD by diluting it ×100 from a solution of 8 M urea (dilution resulting in a final concentration of 80 mM urea) into a buffer, of pH 7.5. Very small (hardly detectable) droplets formed in the absence of salt, whereas 150 mM NaCl apparently accelerated the reaction (Fig. 1b). In a third setup of initiating LLPS, the fused MBP solubility tag of hnRNPA2 LCD-MBP was cleaved by TEV protease. In these conditions, hardly any droplets were seen early on but appeared much later in a practically salt-independent manner (Fig. 1c), which is probably explained by that the rate-limiting step of LLPS in this system is the cleavage reaction itself.

To demonstrate that the full process of phase separation can be monitored upon the pH jump, we followed LLPS by observing the turbidity of the sample at 600 nm (selecting it from a range of wavelengths, Fig. S2). Turbidity of the sample increases rapidly to a maximum within minutes and then decays slowly, reflecting the formation and maturation of droplets by LLPS (Fig. 1d). By spinning down the solutions, the protein was found in the supernatant at pH 11.0, but completely in droplets (without any detectable protein in the supernatant) at pH 7.5. The addition of 150 mM NaCl slowed down the evolution of the turbidity signal, indicating that electrostatics play an important role in the LLPS of the protein (Fig. 1d). In the urea-dilution system, turbidity reached a maximum only after 1 h, then slowly decreased over time (Fig. 1e) and 150 mM NaCl accelerated, rather than decelerated, this reaction, in accord with earlier observations in a similar urea-dilution experiment[11]. This observation would suggest that LLPS of this protein is not primarily driven by electrostatic interactions. A striking difference in LLPS kinetics was also observed in the TEV-cleavage setup. In this case, turbidity of the solution showed a transient minor increase and decrease over time (Fig. 1f), with only negligible effect of salt, suggesting that enzymatic cleavage may be rate-limiting to LLPS in this case. We may conclude that the three approaches lead to very different behaviors that may probably be attributed to limitations caused by either residual denaturant in the system (present in dilution from 8 M urea) or kinetic interference of the enzymatic modification (slow cleavage of a solubility tag).

It is to be noted that the scattering signal in turbidity measurements gives arbitrary units and not units of molarity, i.e., the non-linear interaction of particles with light makes quantitative interpretation and comparison of the kinetic traces difficult. To approach these more directly, we followed the kinetic trace of LLPS by dynamic light scattering (DLS) that enables direct measurement of particle size. In accord with the observations with microscopy, in the case of the pH jump, initially, small droplets of about 300 nm in diameter formed, growing slowly to a maximum of 1500 nm after approximately 1.5 h (Fig. 1g), with an exponential time dependence of $t$ to the power of 1/3, which is characteristic of Ostwald ripening[17].

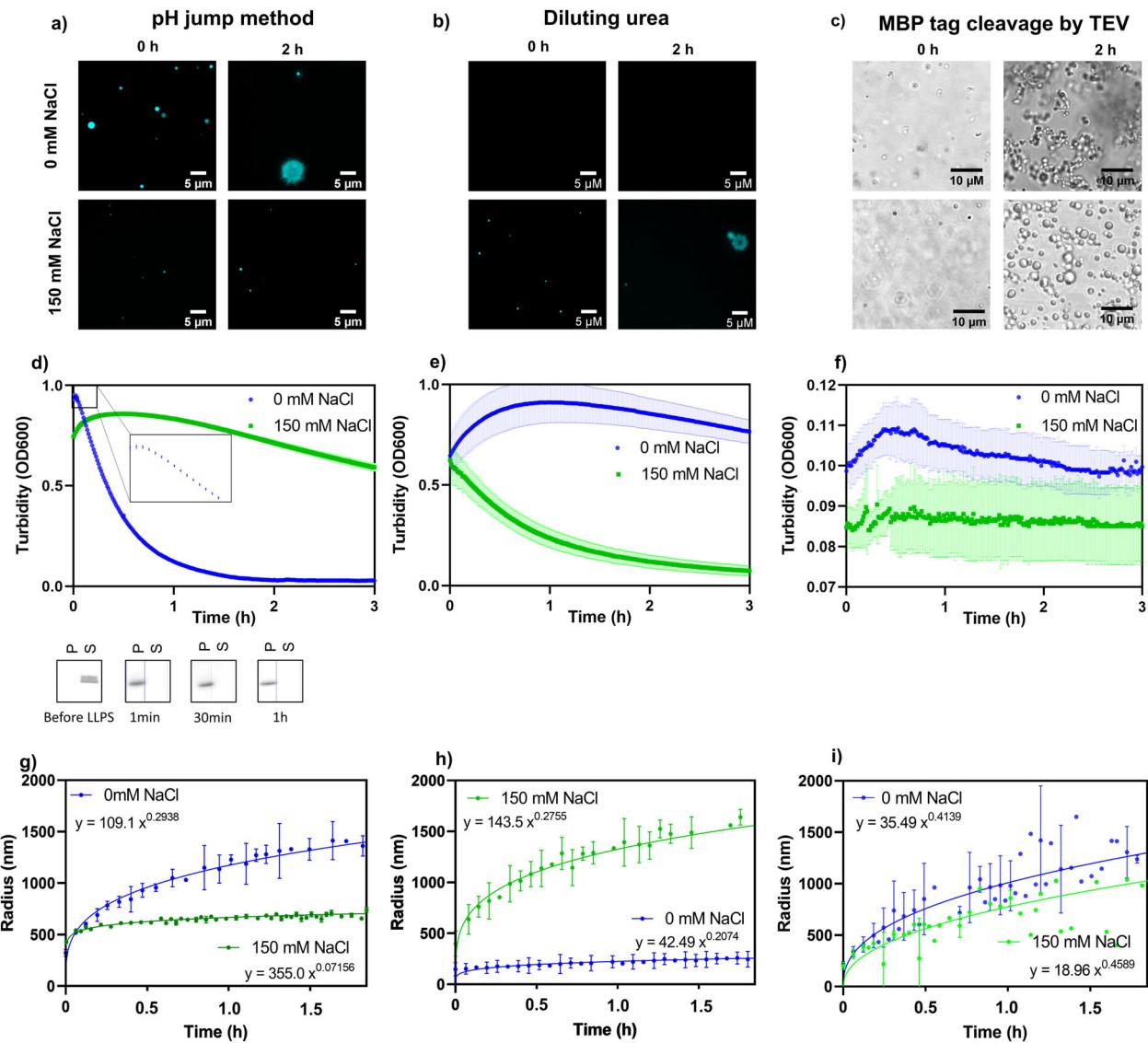

**Fig. 1 Kinetics of phase separation of hnRNPA2 LCD induced by three different approaches.** LLPS of hnRNPA2 LCD (at a final concentration = 20 μM) was initiated by three principally different approaches that result in very different kinetic trajectories: (1) pH jump from pH = 11.0 to 7.5 (panels **a**, **d**, and **g**), (2) ×100 dilution from an 8M-urea solution resulting in a final concentration of 80 mM urea (**b**, **e**, and **h**), and (3) cleavage of the His-tagged MBP of an hnRNPA2 LCD-MBP fusion construct by TEV protease (**c**, **f**, and **i**). LLPS was monitored over time in the absence and presence of 150 mM NaCl, by three different methods (where t = 0 corresponds to the addition of concentrated low-pH buffer in the pH jump, dilution by a large volume of buffer in the urea-dilution experiment and addition of a small volume of concentrated TEV solution for cleavage of MBP tag). First, droplets were visualized by fluorescence microscopy using Dylight 488-labeled LCD (mixed into ×200 excess of non-labeled LCD) by pH jump (**a**) and urea dilution (**b**) immediately after initiating LLPS (marked 0 h) and after 2 h of incubation, in the absence and presence of 150 mM NaCl. Upon TEV cleavage (**c**), we had to use phase-contrast microscopy, as the MBP tag precluded specific labeling of the LCD segment of the construct. Turbidity at 600 nm (OD600) was measured (**d**, **e**, and **f**) without NaCl (blue lines) and with 150 mM NaCl (green lines). An inset in (**d**) shows the initial 15 min of the reaction, whereas gel images below the panel show that the protein is in solution (S) before LLPS whereas it is in the pellet (P) at the different times indicated after phase separation. The size evolution of droplets formed was followed by DLS (**g**, **h**, and **i**), in the absence of NaCl (blue lines), and with 150 mM NaCl (green lines, fitting functions are indicated on the panels). Please note that DLS measurements could not be analyzed past 1.5 h, probably due to aggregation of the protein. All kinetic traces are mean ± SD of experiments in triplicate (n = 3).

Droplet size evolved much slower and reached a maximal value of 600 nm after 1.5 h in the presence of 150 mM NaCl. DLS measurements also confirmed the different kinetic profiles of LLPS upon dilution from urea (Fig. 1h) or MBP cleavage (Fig. 1i). In the presence of urea, again, the effect of salt is reversed, whereas in the MBP-cleavage approach it is hardly observable.

To uncover the reasons for the observed differences and to demonstrate the artificial effects of urea dilution and MBP-tag cleavage, we carried out further experiments. We observed that

80 mM urea added to hnRNPA2 LCD before the pH jump to 7.5 had hardly any effect on the kinetic trace followed by turbidity (Fig. S3a), whereas reaching the same final urea concentration by dilution from an initial solution in 8 M urea had a serious effect (Figs. 1e and S3a). These results suggest that urea at a low (residual) concentration does not much interfere with LLPS, unlike urea at denaturing initial concentrations, from which the protein has to relax to its native state. These effects can be rationalized by the solvation of protein backbone and side chains

by urea, explaining urea-mediated denaturation[18], which may be altered by strong salt-urea interactions[19,20].

We also carried out further analysis of the MBP cleavage process. The addition of TEV protease to the pH-jump system shows that the presence of the protease itself has some effect on LLPS kinetics (Fig. S3b). By following the full trajectory of the cleavage of hnRNPA2 LCD-MBP, more complications appear: cleavage does not proceed to completion, but reaches a plateau after a few hours (Fig. S3c). When we spun down the particles and analyzed the supernatant and the pellet via gel electrophoresis, most of hnRNPA2 LCD was present in the droplets, together with the other components of the mixture, hnRNPA2 LCD-MBP, cleaved MBP tag, and even TEV protease (Fig. S3d). That is, TEV cleavage is rate-limiting and LLPS itself may interfere with it, resulting in a complex kinetic scheme reflecting multiple processes including cleavage, LLPS, and the incorporation of various molecular species in the droplets.

These results underline that the pH jump enables instantaneous induction of LLPS under near-native conditions, whereas other approaches may support complex and artificial kinetic schemes. An important caveat with describing the kinetics of LLPS, however, is that plate reader-based turbidity assays and DLS are not fast enough to resolve early nucleation and growth events in droplet formation, despite rapid transition to LLPS-favoring conditions by the pH jump (Fig. 1d–i). To demonstrate that our novel approach enables to follow pre-steady state kinetics of LLPS, we have carried out stopped-flow experiments with the three systems (pH jump, urea dilution, and MBP tag cleavage) and recorded LLPS within the first 100 ms and 10 s of the reaction (Fig. S4). The three kinetic traces show apparent differences. There is an apparent rapid nucleation of LLPS upon pH jump, which is invisible to urea dilution (in stopped-flow a ×100 dilution of 8 M urea, i.e., a transition from fully LLPS-inhibiting to LLPS-promoting conditions, is technically not even possible) and TEV cleavage (the reaction practically does not even start in the first few milliseconds).

**Phase separation of TDP-43 LCD, NUP98 LCD, and ERD14.** Next, we sought to generalize the pH-jump approach by demonstrating its rational extension to other proteins. First, we noted that the pH-regulated charge state of a protein is a general determinant of its LLPS[21], as observed in many cases, such as lysozyme[22], TDP-43[6], Sup35[14], and Pab1[8]. This effect can be rationalized by scrutinizing a typical phase diagram (Fig. S5a). In general, the high net charge of a protein at an extreme pH ($pH_{SOL}$) is inhibitory to its condensation, an intermediary net charge at native pH ($pH_{LLPS}$) is conducive of LLPS, whereas a net charge close to zero at around its pI would strongly promote its aggregation[3,23,24]. To demonstrate that this principle provides a generic approach for studying the kinetic trajectory of LLPS of other aggregation-prone proteins, we have selected three more candidate proteins, the LCD of TDP-43 and NUP98, and the plant stress protein ERD14 (Fig. S6). We analyzed their net charge-pH curves (Fig. S5b), to select an appropriate pH for each protein at which it remains in solution, and from where the native pH, promoting LLPS, can be reached without crossing the pI of the protein. The net charge of each protein shows a gradual decrease with increasing pH, due to deprotonation of their acidic residues (pKa Asp: 3.65, Glu: 4.25), lysine(s) (pKa: 10.28), and tyrosine(s) (pKa: 10.07). From the curves, we selected pH = 3.5 for TDP-43 LCD, pH = 3.0 for NUP98 LCD and pH = 11.0 for ERD14.

We found that all the proteins are in solution at the respective pH selected, and readily undergo LLPS when their pH is brought to 7.5 (or 6.6, for ERD14) by a small volume of concentrated buffer (Fig. 2). Their LLPS was first visualized by fluorescent microscopy

(Fig. 2a–c), which showed the development of sub-µm sized droplets that grew over time in a salt-dependent manner. We then monitored the kinetics of their phase separation by the turbidity assay at 600 nm (Fig. 2d–f; in the case of ERD14, an RNA analog, poly(U) and a molecular crowder, PEG, had to be added, without which no LLPS occurred, cf. Fig. S7) and supportive DLS (Fig. 2g–i, with an apparent heterogeneity of size distribution on the primary DLS curves, Fig. S8). Importantly, the exponent of particle growth showed quite some deviation from the value 1/3 indicative of Ostwald ripening (ranging from 0.2 to 0.5). We interpret these deviations as a sign of several factors, such as: (i) no clarity of final state, (ii) sedimentation and (iii) wetting of droplets, and (iv) aggregation of the protein, due to which we cannot follow a signal strictly proportional to the number of droplets.

**Native-like conditions allowed by initiating phase separation by a pH jump.** Given the basic differences between the different approaches of initiating LLPS, it is of interest to clarify if it is the pH-jump system that entails the least artificial situation and enables to approach the kinetics of LLPS under the most "native-like" conditions. To underline this view, we first followed the kinetics of the LLPS of all four proteins at different concentrations. The linearity of maximum values of turbidity curves as a function of concentration (Fig. S9) argues for the lack of distortion of kinetic trajectories. We also refer to the general notion that a dominant force driving LLPS is cation-pi interactions between cationic (basic, Arg and Lys) and aromatic (Trp, Phe, and Tyr) residues of proteins[1]. When we scrutinize the sum of cationic and aromatic amino acids in the four proteins (Table S1 and Fig. S10), they only vary between 12.1% and 22.7%, suggesting that cation-pi interactions are important in the LLPS of each protein. However, when we only compare the sum of the two residues that change their charge state at pH = 11.0 (Lys and Tyr, cf. pKa values above), they appear to be very different: very high in hnRNPA2 LCD and ERD14, but very low in TDP-43 LCD and NUP98 LCD. This means that LLPS is basically retarded at pH = 11.0 in the case of the first, but not the second, two proteins, which provides a rationale for the selection of particular $pH_{SOL}$ values of the four proteins. In addition, we have also carried out control experiments with two globular proteins, BSA and lysozyme, which do not function by phase separation, yet they have been suggested to undergo LLPS under extreme conditions[25,26]. By pH jump, however, they show no sign of phase separation (Fig. S11), which again argues that pH jump creates "native-like" conditions for studying LLPS.

**Conclusion**

Kinetics of LLPS encompassing steps of nucleation, growth, and transition(s) between material states liquid, gel, and aggregate, are important, yet largely neglected, aspects of phase separation. Experimental studies of LLPS kinetics are complicated by the aggregation propensity of many of the respective proteins, and the extreme uncertainty in reproducing nucleation-dependent kinetic schemes. Here, we present a generic method that can be used for studying the kinetics of LLPS of any aggregation-prone proteins. We propose that by analyzing its charge-pH curves and cationic/aromatic residue content an appropriate pH can be selected where the protein has a high net charge, it stays stably in solution and its phase separation can be initiated by a change of pH. As the pH jump only requires the addition of a small amount of concentrated buffer, the composition of the system is free to vary and can be accurately controlled, thus providing an adaptable approach that is devoid of artefacts arising from the initial presence of a strong denaturant (e.g., 8 M urea), large dilution effects (e.g., from a solution of high salt), the slow and incomplete cleavage of a solubility tag (e.g., MBP) or the enzymatic

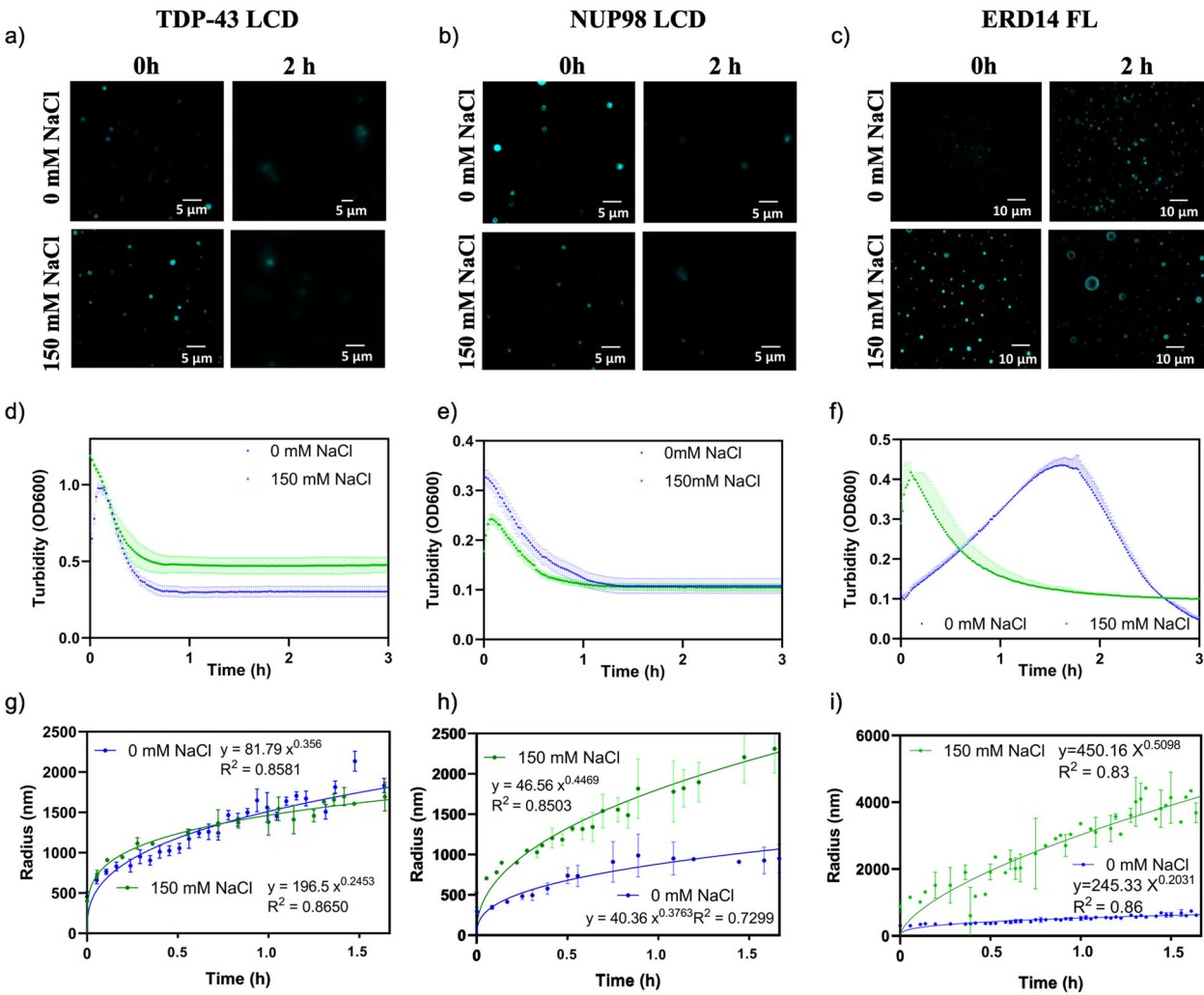

**Fig. 2 Phase separation of three different proteins induced by the pH jump approach.** LLPS of TDP-43 LCD (**a**, **d**, and **g**), NUP98 LCD (**b**, **e**, and **h**) and full-length ERD14 (**c**, **f**, and **i**) was induced by a pH jump from pH = 3.5 to 7.5 (TDP-43 LCD, at 80 μM), pH = 3.0 to 7.5 (NUP98 LCD, at 10 μM) and pH = 11.0 to 6.6 (ERD14, at 20 μM in the presence of 1 mg/ml poly(U) and 8% PEG 6000). Droplets were visualized by fluorescence microscopy of Dylight 488-labeled proteins (mixed into ×200 excess of non-labeled proteins, **a**, **b**, and **c**) without and with salt (150 mM NaCl) immediately after initiating LLPS (marked 0 h) and after 2 h of incubation. LLPS was also monitored by turbidity (OD600) measurement (**d**, **e**, and **f**) in the absence of NaCl (blue lines) and in the presence of 150 mM NaCl (green lines). The size evolution of droplets formed upon LLPS was also followed by DLS (**g**, **h**, and **i**), in the absence of NaCl (blue lines), and in the presence of 150 mM NaCl (green lines, fitting functions are also shown on panels). All kinetic traces are mean ± SD of experiments in triplicate (n = 3).

incorporation of a post-translational modification (e.g., phosphorylation by a kinase). As the change in pH is instantaneous, this approach is amenable for studying the pre-steady state kinetics of nucleation and growth events of LLPS and provides a basis for further, detailed studies to unravel atomistic details of the mechanism of LLPS. Finally, it should also be appreciated that such a pH jump can correspond to the real physiological signal inducing LLPS, as noted in stress-induced acidification and phase separation of stress-sensing proteins Pab1[8] and Sup35[14] or gelation of the entire cytoplasm of yeast cells[24].

## Methods

**Protein production: transformation of bacteria.** Competent *E. coli* BL21 STAR (for protein expression), or NEB5α (for plasmid purification) cells were heat-transformed in Luria broth (LB) medium at 42 °C. After 1 h at 37 °C, the solution was streaked on agar plates with proper antibiotics. After overnight incubation, a single colony was chosen, from which a glycerol stock was prepared and stored at −80 °C. Precultures were prepared by dipping a sterile toothpick in the glycerol stock and incubating it in 50 ml LB with the selectable marker.

**hnRNPA2 LCD expression and purification.** N-terminally polyHis-tagged LCD (region R190 – Y341, Fig. S6) of hnRNPA2 (UniProt P22626), located in a prokaryotic expression vector plasmid pJ411 with a kanamycin resistance gene as a selectable marker (lacI gene) preceded by lac operon (Addgene construct: https://www.addgene.org/98657) was a gift from Prof. N.L. Fawzi (Brown University, Providence, RI, USA).

Preculture was added to 1 L of terrific broth media with kanamycin, and incubated at 37 °C, shaking until an OD600 = 0.6–0.8. Protein expression was induced by adding 1 mM Isopropyl β-D-1-thiogalactopyranoside (IPTG) and the temperature was lowered to 26 °C. After overnight incubation, cells were harvested by centrifugation at 4 °C, 5000 × g for 20 min. The cells were then flash frozen and stored at −80 °C.

Frozen cells were thawed and resuspended in lysis buffer: 20 mM Tris-Cl, 500 mM NaCl, 10 mM imidazole, supplemented with 1 mM dithiothreitol (DTT), 0.1 mM phenylmethylsulfonyl fluoride (PMSF), 0.5 mM benzamidine hydrochloride (BA), and 1 tablet Roche complete EDTA-free protease inhibitor per 50 ml. The cells were lysed by sonication (on a Sonics VCX-70 Vibra cell) for 15 min (5 s pulse on, 5 s pulse off, 70% amplification) on ice to avoid heating the sample. Inclusion bodies, containing hnRNPA2 LCD, were pelleted by centrifugation at 24 000 × g for 1 h at 4 °C. The pellet was resolubilized in a denaturing buffer: 20 mM Tris-HCl, 500 mM NaCl, 10 mM imidazole, 1 mM DTT, 3 M urea, pH 8.0 and centrifuged for 1 h at 24 000 × g and 4 °C to pellet bacterial debris. The supernatant was filtered through a 0.45-μm pore filter and loaded onto

a Nickel-charged IMAC column (HisTrap™ HP – GE Healthcare), to which the polyHis tag of hnRNPA2 LCD binds. Bacterial debris was washed away with a denaturing buffer, after which the bound hnRNPA2 LCD was eluted with a linear imidazole gradient 0–250 mM.

The eluted hnRNPA2 LCD with polyHis tag was cleaved by polyHis-tagged TEV protease in a 50 mM NaH$_2$PO$_3$, 20 mM NaCl, 3 M urea, pH 7.0 buffer at room temperature. After overnight incubation, the cleaved tag and the protease were removed by running the solution over a Nickel-charged IMAC column, collecting the flow-through that contained the cleaved protein. This flow-through was run over a gel-filtration column to remove all contaminants.

Purified hnRNPA2 LCD was extensively dialyzed for 2 × 4 h, and once overnight to 0.01 M CAPS buffer, pH 11.0 applied at a 1:100 volume ratio. The protein was stored at a final concentration of 20 µM at −80 °C. Protein concentration was determined by densitometry of Coomassie-stained SDS-PAGE gels, and by QUBIT®.

For the urea dilution experiments, hnRNPA2 LCD was not dialyzed, but buffer-exchanged to 8 M urea, and concentrated to 2–4 mM.

**hnRNPA2 LCD-MBP expression and purification**. hnRNPA2 LCD-MBP (containing a His tag on the MPB tag) (region R188-Y341, Fig. S6) of hnRNPA2 (UniProt P22626) (Addgene plasmid #98661; http://n2t.net/addgene:98661; RRID: Addgene_98661) was a gift from Prof. N.L. Fawzi (Brown University, Providence, RI, USA).

Precultures were added to LB medium and incubated until they reached an OD600 = 0.6–0.8 at 37 °C, 170 rpm. 1 mM IPTG was added to induce hnRNPA2 LCD-MBP expression. After incubating for 4 h, the cells were harvested by centrifugation at 4 °C, 5000 rpm for 20 min, flash-frozen, and stored at −80 °C.

The cells were thawed and resuspended in lysis buffer: 100 mM KCl, 50 mM HEPES, 0.5 M NaCl, 10 mM Imidazole, pH 7.5 supplemented with 1 mM dithiothreitol, 0.1 mM PMSF, 0.5 mM BA, and 1 tablet Roche complete EDTA-free protease inhibitor per 50 ml on ice. The cells were lysed by sonication (on a Sonics VCX-70 Vibra cell) for 10 min (5 s pulse on, 5 s pulse off; 70% amplification) on ice to avoid heating the sample.

Contaminants were pelleted by centrifugation at 20 000 × g for 1 h at 4 °C. The supernatant was filtered through a 0.45-µm pore filter and loaded onto a Nickel-charged IMAC column (HisTrap™ HP – GE Healthcare) and eluted with a linear imidazole gradient 0–250 mM. The remaining contaminants were removed by gel filtration and the protein was either immediately used or stored for a few days at 4 °C.

**TDP-43 LCD expression and purification**. LCD (region N267-M414, Fig. S6) of TDP-43 (UniProt Q13148) plasmid (Addgene plasmid https://www.addgene.org/98669/) was a gift from Prof. N.L. Fawzi (Brown University, Providence, RI, USA) and it is expressed in E. coli BL21 cells. NZYM media in 1 L bottles were inoculated, and cells were grown at 37 °C until OD600 was between 0.6 and 0.8. Expression was induced by adding 1 mM IPTG, and the cells were grown for 4 h at 37 °C. The cells were harvested by centrifugation at 4 °C, 5000 rpm for 15 min. The cells were frozen and stored at −80 °C. Cell pellet corresponding to 1 L were thawed, and resuspended in 50 ml lysis buffer: 500 mM NaCl, 20 mM Tris HCl, 1 mM DTT, 1 tablet Roche complete EDTA-free protease inhibitor, 0.5 mM BA, 0.1 mM PMSF, pH 8.0 and lysed by sonication on ice (15 min, 70% amplitude, 10 s on, 10 s off).

Inclusion bodies were pelleted by centrifugation at 5000 × g for 5 min at 4 °C. The pellet was resolubilized in denaturing buffer (20 mM Tris HCl, 8 M urea, 500 mM NaCl, 10 mM imidazole, 1 mM DTT, pH 8.0) and sonicated on ice for 5 min (cycles of 5 s on and 5 s off at 70% amplitude). The solution was centrifuged at 20,000 × g for 1 h at 4 °C. The supernatant was filtered through a 0.45-µm pore filter and loaded onto a Nickel-charged IMAC column (HisTrap™ HP – GE Healthcare). TDP-43 LCD was eluted with a linear imidazole gradient of 0–500 mM. Protein fractions were combined and pH was adjusted to 7.0.

TEV protease (1:100 w/w) was added to cleave the poly-His tag overnight at room temperature. The cleaved tag and the protease were removed by running the solution over a Nickel-charged IMAC column, and the flow-through buffer was exchanged into 20 mM Tris-HCl buffer pH 3.5. The protein was filtered through a 0.22-µm pore filter and stored at a final concentration of 55 µM at −80 °C. Protein concentration was determined by densitometry of Coomassie-stained SDS-PAGE gels and by QUBIT®.

**NUP98 LCD expression and purification**. The plasmid of the NUP98 LCD (region M1-Q407, Fig. S6) of Nup98 (UniProt P52948) provided by Addgene (Addgene plasmid # 38037; http://n2t.net/addgene:38037; RRID: Addgene_38037) was a gift from Prof. Roderick Lim[27]. The protein was expressed in E. coli BL21 cells. Protein in inclusion bodies was extracted and solubilized in a buffer of 8 M urea, 100 mM Na2HPO4, 10 mM DTT, and 10 mM Tris-HCl, pH 8.5, purified through Ni-NTA affinity chromatography. The eluted protein was finally dialyzed into PBS buffer, pH 3.0 and kept frozen at −20 °C

**ERD14 expression and purification**. ERD14 (full-length, UniProt P42763, Fig. S6) was purified as described in ref. [28]. Briefly, ERD14 expressing BL21(DE3) cells were collected after induction with 0.7 mM IPTG at 30 °C overnight. Cells were then lysed in lysis buffer containing 50 mM Tris pH 8.0, 150 mM NaCl, 1 mM BA, 0.5 mM PMSF, 5 mg DNase, 20 mM MgCl$_2$ and protease inhibitor cocktail (Roche), and sonicated for 3 min (10 s pulse on, 10 s pulse off, 60% amplitude) using Sonics Vibra Cell. After centrifugation (20 min, 20000 × g, and 4 °C), the supernatant was boiled (20 min) to remove contaminating proteins. After centrifugation at high speed again, the lysate was desalted using HiPrep™ 26/10 desalting column equilibrated with 50 mM Tris pH 8.0, 0.05 mM BA, 0.05 mM PMSF buffer. The collected fractions were pooled and loaded onto a HiTrap DEAE Sepharose FF column (GE Healthcare Life Sciences). Following washing with Tris buffer, ERD14 was eluted with a step gradient of 10 to 500 mM NaCl in buffer. The purification was repeated one more time with MonoQ 4.6–100 column to remove all contaminants. The samples were analyzed on SDS-PAGE gel for purity and stored at −20 °C.

**Globular control proteins**. Control proteins BSA (VWR Life Science in a lyophilized form) and lysozyme (Thermo Scientific) were from commercial sources. They were dissolved in 10 mM CAPS buffer, pH 11.0 (BSA), and 20 mM MES buffer, pH 5.5 (lysozyme) prior to LLPS experiments.

**Initiating phase separation**

*hnRNPA2 LCD*. hnRNPA2 LCD stays in solution in 10 mM CAPS buffer at pH 11.0. Its immediate phase separation can be induced by decreasing its pH to 7.5. This pH drop is achieved by adding an appropriate amount tested for the actual batch of protein (typically, 2% V/V) of 0.5 M MES, pH 5.5.

Alternatively, phase separation was induced by diluting the 8M-urea solution 100x (to a final concentration of 80 mM), in 20 mM HEPES pH 7.5 buffer.

*hnRNPA2 LCD-MBP*. Phase separation was induced by adding TEV to cleave off the MBP tag at an hnRNPA2:TEV molar ratio of 100:1 (upon which the His-tag remains attached to MBP).

*TDP-43 LCD*. TDP-43 LCD is stable in 20 mM MES buffer at pH 3.5 and phase separates when its pH is increased to 7.5 with an appropriate volume of 0.5 M CAPS buffer, pH 11.0. Usually, for 200 µl of TDP 43 LCD (55 µM) we added 5 µl of CAPS buffer.

*NUP98 LCD*. NUP98 LCD is stable in a 50 mM PBS buffer at pH 3.0 and phase separates when its pH is increased to 7.5 in the presence of 10% PEG (Mw = 4000 g/mol). Therefore, the protein was stored under these conditions and its LLPS was induced by applying an appropriate volume of 0.1 M CAPS buffer, pH 11.0.

*ERD14*. ERD14 is stable in 50 mM Tris buffer at pH 11.0 and phase separates when its pH is decreased to 6.6. The protein was normally stored in lyophilized form and dissolved in a suitable buffer before the assay. The phase separation was induced by adding an appropriate amount of HCl to get the pH to 6.6 in the presence of 1 mg/ml poly(U) and 8% PEG 6000. Normally, for 200 µl of ERD14, we added 5 µl of 0.1 M HCl.

*Globular control proteins*. pH jump of BSA (dissolved in 10 mM CAPS buffer, pH 11.0) was initiated by the addition of a small volume of 0.5 M MES buffer, pH 5.5. pH jump of lysozyme (dissolved in 20 mM MES buffer, pH 5.5) was initiated by the addition of an appropriate volume of 0.5 M CAPS buffer, pH 11.0.

**Fluorescent labeling of proteins**. Proteins were labeled for fluorescence microscopy by the fluorescent Dylight® 488 dye (Thermo Scientific) as described below. Labeling protocols were set to result in a typical labeling stoichiometry of 0.5 molecule dye/protein.

*hnRNPA2 LCD*. 100 µl of 8 mg/ml hnRNPA2 LCD in TEV cleavage buffer (3 M urea, 200 mM NaCl, 50 mM NaH$_2$PO$_4$) was dialyzed against 0.1 M sodium carbonate buffer, pH 8.5. 10 mg/ml of Dylight® 488 was dissolved in DMSO and added to the protein at a final concentration of 0.05 mg/ml. The solution was incubated at room temperature for 1 h and then dialyzed against 0.01 M CAPS, pH 11.0 storage buffer, to remove the excess of unbound fluorophore. Fluorescently labeled hnRNPA2 LCD was protected from light and stored at −80 °C.

*TDP-43 LCD*. 100 µM of Dylight® 488 was used to label TDP-43 LCD for fluorescent microscopy measurements. TDP 43 LCD (55 µM) in 1x PBS was added to the dye and incubated at room temperature for 30 min in darkness. The mixture was then desalted with the Zeba™ Desalting Spincolumns, 7 K MWCO (Thermo Fisher Scientific) to the reaction buffer (1x PBS, 10 mM MgCl$_2$, 0.05% Tween 20). Fluorescently labeled TDP-43 LCD was protected from light and stored at −80 °C.

*NUP98 LCD*. 100 µl of 100 µM NUP98 LCD in PBS buffer was labeled with Dylight® 488 at a molar ratio of 1:3. The solution was incubated at room temperature for 1 h and then dialyzed against 50 mM PBS buffer, pH 3.5 (storage buffer), to remove the excess of unbound fluorophore.

ERD14. To prepare the sample for microscopic imaging, 100 μM ERD14 in 1x PBS was labeled with Dylight® 488 dye (1:3 molar ratio). Desalting step to 1x PBS buffer was applied as described for TDP-43 LCD, to remove the excess dye.

**Turbidity measurements**. To measure the turbidity of solutions, non-binding black 96 well plates of transparent bottom (Greiner bio-one, chimney well, μclear®) were used. The solutions were mixed and the plate was covered with a transparent film (VIEWsealTM). The absorbance of the solution was measured at 600 nm for 6 h on a BioTek SynergyTM Mx plate reader at 25 °C, with continuous shaking. The experiments were conducted in triplicate and mean ± SD of measured values were calculated.

**Dynamic light scattering measurements**. Dynamic light scattering (DLS) measurements were carried out on a DynaPro NanoStar (Wyatt) instrument. A disposable cuvette (WYATT technology) was filled with 100 μl of protein solution at pH and concentration values at which LLPS occurred. The sides of the cuvette were filled with water and a cap was put on top. The intensity of scattered light was recorded at a scattering angle of 95° at 25 °C, for a period of 6 h, collecting 10 acquisitions (8 s each). Each measurement was repeated at least 3 times. The software package DYNAMICS 7.1.9 was used to analyze the data and to calculate particle size by the following formulas describing $G$ autocorrelation function (Eq. (1)) and $D_t$ translational diffusion coefficient (Eq. (2)):

$$G^2(\tau) = <I>^2\left(1 + \alpha e^{-2D_t q^2 \tau}\right) \tag{1}$$

$$D_t = \frac{kT}{6\pi\eta R_h} \tag{2}$$

Where $\tau$ is delay time, $q$ is refractive index, $R_h$ is hydrodynamic radius, $k$ is the Boltzmann's constant, $T$ is temperature, $\eta$ is viscosity, and $I$ is intensity.

**Fluorescent and brightfield microscopy**. Microscopy measurements were carried out on a Leica DMi8 microscope equipped with a Leica DFC7000 GT camera. Dylight® 488-labeled proteins were each mixed with ×200 excess of the same, non-labeled, protein. Phase separation was then induced by changing the pH of the protein solution as described earlier. The solution was incubated at 25 °C and droplets were visualized with ×100 oil-immersion objectives with brightfield, and/or fluorescence microscopy (applying a FITC filter).

**Stopped-flow measurements**. To follow the early stages of phase separation, kinetic experiments were performed on a stopped-flow instrument (Applied Photophysics stopped-flow SX20), by monitoring the change in absorbance at 600 nm. The reaction was recorded immediately after (following an instrument-specific deadtime of about 1 ms) the protein sample in the first syringe (A) and LLPS-inducing buffer in the second syringe (B) were injected into the measuring chamber simultaneously at a volume ratio 1:1. OD600 of sample buffer was used as control. Experiments were set up to reach the same final protein concentration (20 μM hnRNPA2 LCD) as in the plate reader-based assays (cf. Fig. 1). Due to the different volume ratios of mixing, different phase-separating buffer combinations were used, however, to reach conditions commensurable with the plate reader-based assays. pH jump: 20 mM MES pH 5.5 was added in chamber B. Urea dilution: hnRNPA2 LCD dialyzed into 160 mM urea was filled in chamber A. TEV cleavage: hnRNPA2 LCD was applied at 40 μM in 20 mM HEPES pH 7.5 in chamber A, and TEV at a concentration of 0.4 μM in chamber B, to reach an hnRNPA2 LCD:TEV molar ratio of 100:1 after dilution.

**Statistics and Reproducibility**. Statistical analysis was performed using GraphPad Prism 5 software. The data are presented as mean ± SD. For statistical comparisons between kinetic traces measured by DLS, a fitting with an exponential function was carried out. Experiments were repeated three times by three different protein preparations (representing biological replicates). Sample size (n) for each experiment appears in figure legend.

**Reporting summary**. Further information on research design is available in the Nature Research Reporting Summary linked to this article.

## Data availability

Raw data for graphs presented in the main figures are available in Supplementary Data 1 (Fig. 1) and Supplementary Data 2 (Fig. 2). All other data are available within the manuscript files or from the corresponding author and first author upon reasonable request.

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

## Acknowledgements

This work was supported by an FWO PhD fellowship in fundamental research, a VUB Spearhead grant (SRP51), grants K124670, K131702, K125340, and FK128133 from the National Research, Development and Innovation Office of Hungary (NKFIH), a

PREMIUM-2017-48 grant from the Hungarian Academy of Sciences, and a PhD Scholarship from the Vietnamese government (911). We also thank Dr. Denes Kovacs (UCB Pharma, Belgium) for his advice on the pH jump approach and Prof. N.L. Fawzi (Brown University, Providence, RI, USA) for providing several constructs to this study.

## Author contributions

J.V.L. purified protein and performed experiments with hnRNPA2 LCD; A.T. purified NUP98; P.N.N. purified and carried out stopped-flow experiments and experiments with ERD14; A.B.S. purified TDP-43 LCD, performed experiments with TDP-43 LCD and NUP98 LCD and analyzed data; D.P. performed data analysis measurements; R.P. carried out sequence analysis of proteins; L.F.D.-A. carried out stopped-flow experiments and analyzed data; P.T., L.V.D.B., and D.M. contributed to the initial design of the experiments; A.B.S. and P.T. wrote the manuscript and all authors were involved in revising it critically for important intellectual content.

## Competing interests

The authors declare no competing interests.
