## [Peer Review File · Communications Biology]

Reviewers' comments:

Reviewer #1 (Remarks to the Author):

This paper by Van Lindt et al presents a new approach that allows for kinetic studies on phase-separating protein in vitro. Most studies in the literature use methods that involve solubility tags, denaturing conditions, or crowding agents and absurd salt conditions. This makes it difficult to interpret results. The method described by Van Lindt et al uses instantaneous pH changes that allow for observation of LLPS in physiologic conditions.

I agree with the authors that current methods are flawed and don't allow for observation of the most interesting dynamic effects, in particular nucleation and growth events. The presented method is simple and has the potential to be used by many groups in the growing LLPS field. Overall, the manuscript is well written and easy to follow, and I recommend publication if the following points can be addressed.

- The alleged rapid increase in turbidity is not clear from the no NaCl condition in Figure 1A. I suggest an inset with a magnification of this part.
- The gel bands in Figure 1A are confusing. They bands appear to be in the pellet at all time points. The text and figure legend states the protein was in the supernatant at pH11 (aka t=0) but this is not shown. Do the insets correspond to a particular time on the x-axis?
- It would be helpful to state in the caption what happens at t=0 in the three different cases (addition of enzyme, sample dilution, buffer addition).
- It is very unclear to me what the arrows in Figure 1G mean.
- All droplets in figure 1H appear oval, and there appear to be insets in the figure. Can the authors comment?
- It would be helpful if the authors could comment on the correlation between their findings with DLS and fluorescence microscopy. Do they find the same overall size with both methods? What if the proteins adopt non-spherical shapes, such as in Figure 2H?
- The authors show that the tag-cleavage system is very complex and all components end up in aggregates. A good control would be to add the enzyme to the WT protein and see if it affects LLPS.
- The authors claim all growth shows a time dependence of $1/3$, as expected for Ostwald ripening. This is a good result, but for some cases, the exponent is not really close to $1/3$, for example 0.20-0.50 in figure 2F. It would be helpful if a confidence interval was shown on these fits, and the authors should consider modifying the strength of this statement.
- Figure S1C: It is unclear what is meant by the comment "Please note that hnRNPA2 LCD stains bad by Coomassie, but is missing from the supernatant.". It appears as if the hnRNPA2 LCD is present in the supernatant, but not in the pellet.

Reviewer #2 (Remarks to the Author):

Lindt, et al describe a method approach to induce liquid phase separation of polypeptides fragments with prion-like characteristics. In their manuscript they demonstrate that the polypeptides can be kept in solution at high pH values and that liquid-like structures can be formed by changing the pH of the reaction solution to more neutral / physiological pH values. They compare this approach to other approaches used in the field, namely the dilution of the protein stored in high concentration of denaturant, such as urea or the enzymatic proteolytic removal of a solubility tag from the polypeptide of interest. They convincingly demonstrate that each approach results in a different behavior when comparing the reactions by turbidity, dynamic light scattering and microscopy measurements. After establishing this framework the authors propose that keeping the polypeptides soluble at pH values

significantly above the isoelectric point of the polypeptide and that reducing the pH value allows the formation of liquid-like structures is a general behavior. To validate this, the authors provide evidence that this approach holds true for additional polypeptides, supporting the notion that a sudden change in the pH of the reaction solution may be a method to study the kinetics of liquid liquid phase separation.

Overall, the study provides an interesting observation and highlights the importance of controlling the conditions that are being chosen to study the problem of interest. In this case the problem was how to start and characterize the reaction that leads to the formation of liquid-like structures. The authors very clearly demonstrate that the storage conditions of the test polypeptide, as well as the test conditions and the method of choice to induce a transition from one to the other state, matters significantly and biases the conclusions. I generally appreciate the approach and find that this information is of general importance to the field. However, clarity, with respect to text, as well as the experimental conditions are missing and should be improved before publication.

My major concern lays in the use of the term "kinetics". The title and through the manuscript the use of this term suggests that using the approach and the methods applied here, allow quantitative description of microscopic and macroscopic rate constants related to liquid liquid phase separation. However, neither the data presented here, nor the methods applied provide quantitative units to conclude about reaction rates. E.g. when using light scattering, the signal is already increased at t_0 , demonstrating that a significant part of the initial reactions occurred within the dead time of the experiment. Moreover, the signals do not relate to quantitative units of either molecular amounts or particles sizes. The authors address this in part by using dynamic light scattering, however, given the sizes of the particles formed, the reaction enters into a regime in which the linearity between mass and scattering intensity no longer hold true and would require Mie theory of intensity scattering. Given the size and nature of the particles one would also expect sedimentation, and possibly also wetting to occur during the time course of the experiment. The scattering intensities would thus not be uniform throughout the experiment, suggesting that only a part of the reaction is described. I thus highly recommend rephrasing the content such that the approach of changing the pH of the reaction solution, possibly in combination with rapid mixing technologies, may enable kinetic studies in the future. Accordingly, I suggest rephrasing the title and the text through out to clearly distinguish the individual claims that can be drawn from the experiments carried out here and what this may enable in the future.

Another main issue is that the experimental conditions are unclear. E.g. I did not find the protein concentrations used in the individual experiments and thus it remains ambiguous whether a comparison among different experiments and methods is possible. In the same line, the author should make clear that the comparison among the three methods differs in more than just one aspect, e.g. the protein construct that was used. While the constructs are clearly stated in the methods section, in the current version of the manuscript the reader is left with the impression that the same construct is used. While I would prefer a comparison in which all parameters are kept constant, except the methodology to induce the transition, I understand that this manuscript focuses in the first part on the comparison and approaches that have been used in the field previously. I appreciate that the authors have tested one and the same protein, under the same condition with and without residual urea. This is highly informative and demonstrates the importance of precise control and description of the experiment as the outcome may have substantial influence on the interpretation.

I believe that after removing the emphasis of a kinetic approach and improving the clarity this manuscript will be of interest to the readers and provides a good resource to study the problem of liquid phase separation in the future.

Minor:

1. I suggest reorganizing the results to show the microscopy results first. In its current form the reader is left unclear about which kind of structure are being formed with which particular approach (pH jump vs. urea-dilution vs. MBP-cleavage). Only later the reader learns that all these treatments results in similar structures, but possibly with different viscoelastic properties. Putting this result first

would sharpen the comparative analysis and help the authors to avoid statements like “, but in some condensed form” (page 3).

2. Page 2, first paragraph: “In vitro, stress-granule proteins, such as TAR DNA-binding Protein 43 (TDP-43), Fused in Sarcoma (FUS) and Heterogeneous Nuclear Ribonucleoprotein A2/B1 (hnRNPA2/B1) form liquid droplets, then undergo gelation and may be converted into aggregated fibrils” This sentence needs a reference.

3. “First, we followed the evolution of the droplets over time by dynamic light scattering (DLS). In the case of the pH jump, initially small droplets of about 300 nm in diameter formed, growing slowly to a maximum of 1500 nm after approximately 1.5 h (Figure 1d), with an exponential time dependence of t to the power of $1/3$, as expected for particle growth by Ostwald ripening 15.” Given the size and nature of the structures I wonder if the behavior observed is truly due to Ostwald ripening (sedimentation, wetting, etc).

4. How do the authors explain the following controversy: “Upon MBP cleavage, hardly any droplets are seen early on but appear much later in a practically salt-independent manner”. However, in their comparative analysis with the other constructs the formation of the structures is strongly salt-dependent. Would this not suggest that the properties are different and/or that the mechanism of structure formation is different?

5. “These results underline that the pH jump enables instantaneous induction of LLPS under near-native conditions, whereas other approaches may support complex and artificial kinetic schemes. Therefore, we sought to generalize this approach by demonstrating its rational extension to other proteins.”

6. The authors may want to emphasize that even for pH jump experiments the used constructs was purified under (mild) denaturing conditions using 3 M urea.

7. Page 8: hnRNPA2 LCD – The title of this section is cryptic and should be replaced with something more appropriate

8. hnRNPA2 LC-MBP: The purification protocol indicates purification via a His-Tag. Is the HisTag present in the experiments?

9. The authors may want to state the labeling efficiency for fluorescently labeled proteins in the methods section and at which ratios the proteins were used.

10. The information about the 96 well plates is missing. This information would be useful.

11. The authors may want to consider that the problem of using absorbance measurements at OD 600 particles results in a detection limit for significant but small particle sizes.

12. The method section about microscopy is incomplete and should e.g. list the objectives used with magnification, numerical aperture, etc.

13. I think the authors should include unrelated polypeptides, such as BSA or alpha Lactalbumin in the pH-jump approach and compare the results to the ones obtained for disordered polypeptides.

14. For Nup98 the microscopy rather suggests that the protein aggregates rather than forms liquid-like structures. How do the authors envision the growth observed by DLS? Why is the image at 2 hours so dim?

15. For Erd14 the authors should include controls with and without polyU, as well as polyU only. The authors should also mention more accurately the conditions, e.g. that this was in the presence of polyU.

16. The authors should test for the ability to dissociate the structures to test for reversibility.

Reviewer #3 (Remarks to the Author):

The communication from Van Lindt, et. al. shows that extreme pH values can be used to solubilize various low complexity domain proteins which have been demonstrated to undergo liquid-liquid phase separation (LLPS) under biochemical conditions which mimic native solution conditions. They have

used this to jump proteins into physiological pH and demonstrated that the kinetics of phase separation is dramatically different between three major methods for inducing phase separation: jump in pH, sudden drop in denaturant (Urea, Guanidinium chloride), or cleavage of a solubilizing tag. The authors argue that the kinetics of phase separation as shown by a pH induced jump is the most "native-like" path toward phase separation given the milder conditions provided by a change in pH. The authors report on an interesting use of a phenomenon that was demonstrated earlier in the literature, albeit for a different purpose, and this work should encourage the use of pH jumps to study phase separation. The experiments support the central argument of different kinetic mechanisms of phase separation induced by different means of inducing phase separation. This reviewer cautions against drawing a quantitative value from the DLS scattering curves. The power law fit of the curves isn't consistent all the time (although this might be due to other competing mechanisms such as aggregation) and detracts from the main point that a significantly different particle distribution exists (also showing some representative curves in the supplement would help hammer this point home as well). Secondly, this reviewer believes emphasis on a jump in pH being a "near-native" like mechanism compared to Urea should be curtailed. Such a jump from either pH 11 or 3.5 to neutral may appear to behave more "native-like" for the proteins tested here, there's no such guarantee this holds for other IDRs which undergo LLPS, and certainly won't for proteins which also contain folded domains. Rather an emphasis on how pH provides an orthogonal mechanism to salt or denaturant to study the molecular mechanism of LLPS would be much more appealing for specialists in the biophysics of LLPS as well the broader audience interested in membraneless organelles.

Communications Biology, Rebuttal

Aug. 24, 2020

Detailed point-by-point rebuttal to reviewer comments on our manuscript:

A generic approach to study the kinetics of liquid-liquid phase separation under near-native conditions

Reviewer #1

This paper by Van Lindt et al presents a new approach that allows for kinetic studies on phase-separating protein in vitro. Most studies in the literature use methods that involve solubility tags, denaturing conditions, or crowding agents and absurd salt conditions. This makes it difficult to interpret results. The method described by Van Lindt et al uses instantaneous pH changes that allow for observation of LLPS in physiologic conditions.

I agree with the authors that current methods are flawed and don't allow for observation of the most interesting dynamic effects, in particular nucleation and growth events. The presented method is simple and has the potential to be used by many groups in the growing LLPS field. Overall, the manuscript is well written and easy to follow, and I recommend publication if the following points can be addressed.

- The alleged rapid increase in turbidity is not clear from the no NaCl condition in Figure 1A. I suggest an inset with a magnification of this part.

> Response: we thank the reviewer for this point, which seems simple but actually highlights the limitation of approaching LLPS by the entire field. We have updated Figure 1a (now Figure 1d) by a small inset to show the initial phase of the curves. However, as the rising phase of the kinetic curve cannot be resolved by the given plate reader-based technique, we have carried out new experiments by stopped-flow, to demonstrate that pH jump enables us to follow kinetics even by a millisecond time resolution. In the new Figure S4, we demonstrate severe differences in early kinetics of the LLPS of hnRNPA2 LCD by the three approaches (pH jump, urea-dilution and TEV cleavage), the implications of which we now discuss in the text.

- The gel bands in Figure 1A are confusing. They bands appear to be in the pellet at all time points. The text and figure legend states the protein was in the supernatant at pH11 (aka t=0) but this is not shown. Do the insets correspond to a particular time on the x-axis?

> Response: thank you, the first gel pair (at pH=11.0, before LLPS) was actually missing from the figure, and is now presented in the updated Figure 1d.

- It would be helpful to state in the caption what happens at t=0 in the three different cases (addition of enzyme, sample dilution, buffer addition).

> Response: we have added the appropriate explanatory sentence to the legend of Figure 1 ("where t = 0 corresponds to the addition of concentrated low-pH buffer in the pH jump, dilution by a large volume of buffer in the urea-dilution experiment and addition of a small volume of concentrated TEV solution for cleavage of MBP tag").

- It is very unclear to me what the arrows in Figure 1G mean.

> *Response: arrows were pointing to small insets in the figure, but have been removed from the updated figure (now Figure 1a) to avoid confusion.*

- All droplets in figure 1H appear oval, and there appear to be insets in the figure. Can the authors comment?

> *Response: There were some technical issues here, so Figure 1H (now Figure 1b) has been replaced and all droplets now appear as spheres.*

- It would be helpful if the authors could comment on the correlation between their findings with DLS and fluorescence microscopy. Do they find the same overall size with both methods? What if the proteins adopt non-spherical shapes, such as in Figure 2H?

> *Response: Please note that (upon the request of another Reviewer) Figure 1 has been significantly rearranged, now begins with microscopic images. Nevertheless, now we provide a more elaborate explanation for the correlation of microscopic and DLS observations (page 4, top)*

- The authors show that the tag-cleavage system is very complex and all components end up in aggregates. A good control would be to add the enzyme to the WT protein and see if it affects LLPS.

> *Response: thanks for noticing this, the labels “pellet” and “supernatant” were actually swapped in Figure S1c, which now has been corrected (now Figure S1d). Further, we have carried out this interesting control experiment, which is now incorporated in the same figure, as Figure S1b: the addition of TEV to the pH-jump experiment does have an effect on kinetics.*

- The authors claim all growth shows a time dependence of $1/3$, as expected for Ostwald ripening. This is a good result, but for some cases, the exponent is not really close to $1/3$, for example 0.20-0.50 in figure 2F. It would be helpful if a confidence interval was shown on these fits, and the authors should consider modifying the strength of this statement.

> *Response: this is a reasonable point (also raised by another Reviewer), as the exponent of the fit depends on many factors, and would only be $1/3$ if Ostwald ripening occurred and we followed a signal strictly proportional to the number of droplets. As noted, however, there is quite some variation in the determined value (from 0.2 to 0.5), which we attribute to factors: (i) no clarity of final state, (ii) aggregation, (iii) sedimentation, and (iv) wetting. In accord, we have toned down our claim that we always encounter Ostwald ripening, and we go at length now in discussing these other factors that contribute to the signal observed (page 5, middle).*

- Figure S1C: It is unclear what is meant by the comment “Please note that hnRNPA2 LCD stains bad by Coomassie, but is missing from the supernatant.”. It appears as if the hnRNPA2 LCD is present in the supernatant, but not in the pellet.

> *Response: as stated above, the bands (“pellet” vs. “supernatant”) were mislabeled, and now we present the correct figure (updated Figure S1d).*

Reviewer #2

Lindt, et al describe a method approach to induce liquid phase separation of polypeptides fragments with prion-like characteristics. In their manuscript they demonstrate that the polypeptides can be kept in solution at high pH values and that liquid-like structures can be formed by changing the pH of the reaction solution to more neutral / physiological pH values. They compare this approach to other approaches used in the field, namely the dilution of the protein stored in high concentration of denaturant, such as urea or the enzymatic proteolytic removal of a solubility tag from the polypeptide of interest. They convincingly demonstrate that each approach results in a different behavior when comparing the reactions by turbidity, dynamic light scattering and microscopy measurements. After establishing this framework the authors propose that keeping the polypeptides soluble at pH values significantly above the isoelectric point of the polypeptide and that reducing the pH value allows the formation of liquid-like structures is a general behavior. To validate this, the authors provide evidence that this approach holds true for additional polypeptides, supporting the notion that a sudden change in the pH of the reaction solution may be a method to study the kinetics of liquid-liquid phase separation.

Overall, the study provides an interesting observation and highlights the importance of controlling the conditions that are being chosen to study the problem of interest. In this case the problem was how to start and characterize the reaction that leads to the formation of liquid-like structures. The authors very clearly demonstrate that the storage conditions of the test polypeptide, as well as the test conditions and the method of choice to induce a transition from one to the other state, matters significantly and biases the conclusions. I generally appreciate the approach and find that this information is of general importance to the field. However, clarity, with respect to text, as well as the experimental conditions are missing and should be improved before publication.

My major concern lies in the use of the term “kinetics”. The title and through the manuscript the use of this term suggests that using the approach and the methods applied here, allow quantitative description of microscopic and macroscopic rate constants related to liquid liquid phase separation. However, neither the data presented here, nor the methods applied provide quantitative units to conclude about reaction rates. E.g. when using light scattering, the signal is already increased at t_0 , demonstrating that a significant part of the initial reactions occurred within the dead time of the experiment. Moreover, the signals do not relate to quantitative units of either molecular amounts or particles sizes. The authors address this in part by using dynamic light scattering, however, given the sizes of the particles formed, the reaction enters into a regime in which the linearity between mass and scattering intensity no longer hold true and would require Mie theory of intensity scattering. Given the size and nature of the particles one would also expect sedimentation, and possibly also wetting to occur during the time course of the experiment. The scattering intensities would thus not be uniform throughout the experiment, suggesting that only a part of the reaction is described. I thus highly recommend rephrasing the content such that the approach of changing the pH of the reaction solution, possibly in combination with rapid mixing technologies, may enable kinetic studies in the future. Accordingly, I suggest rephrasing the title and the text throughout to clearly distinguish the individual claims that can be drawn from the experiments carried out here and what this may enable in the future.

Another main issue is that the experimental conditions are unclear. E.g. I did not find the protein concentrations used in the individual experiments and thus it remains ambiguous

whether a comparison among different experiments and methods is possible. In the same line, the author should make clear that the comparison among the three methods differs in more than just one aspect, e.g. the protein construct that was used. While the constructs are clearly stated in the methods section, in the current version of the manuscript the reader is left with the impression that the same construct is used. While I would prefer a comparison in which all parameters are kept constant, except the methodology to induce the transition, I understand that this manuscript focuses in the first part on the comparison and approaches that have been used in the field previously. I appreciate that the authors have tested one and the same protein, under the same condition with and without residual urea. This

is highly informative and demonstrates the importance of precise control and description of the experiment as the outcome may have substantial influence on the interpretation.

I believe that after removing the emphasis of a kinetic approach and improving the clarity this manuscript will be of interest to the readers and provides a good resource to study the problem of liquid phase separation in the future.

> Response: these are important points, and we have taken great care to contain them:

(i) for the issue of kinetics, we appreciate the point that kinetics can only be measured if we record signals that are proportional to molar values, thus real rate constants can be determined. Further, we take it that rapid mixing and pre-steady state kinetics are essential to describe nucleation and growth events of LLPS. We have responded to these points by carrying out stopped-flow experiments with the three systems (new Figure S4), and compare real kinetics in the first 100 ms and 10 s of the reactions (by pH jump, urea-dilution and TEV cleavage). As this experiment was central to the suggestion, we feel we are now justified in claiming that we can approach the “kinetics” of LLPS, thus we have kept this claim in the title but adapted the text at many points accordingly (e.g. page 4, bottom).

(ii) we completely agree that experimental conditions were not clearly described, and we now indicate in the legends of respective figures the exact concentrations used.

(iii) We now emphasize in the text better that the constructs of the different approaches are in fact the same. Please note that for further clarification in this direction, we have carried out another control experiment, in which we added TEV protease to the pH jump system of hnRNPA2 LCD (new panel: Figure S1b).

Minor:

1. I suggest reorganizing the results to show the microscopy results first. In its current form the reader is left unclear about which kind of structure are being formed with which particular approach (pH jump vs. urea-dilution vs. MBP-cleavage). Only later the reader learns that all these treatments result in similar structures, but possibly with different viscoelastic properties. Putting this result first would sharpen the comparative analysis and help the authors to avoid statements like “, but in some condensed form” (page 3).

> Response: thank you for this suggestion, we have rearranged the order of presenting results and adapted Figure 1 and Figure 2, to start with microscopic images (due to which

the entire story-line changed, cf. page 3 middle and page 4, top). Further, we can now claim we observe “droplets” and not “some condensed form” of material.

2. Page 2, first paragraph: “In vitro, stress-granule proteins, such as TAR DNA-binding Protein 43 (TDP-43), Fused in Sarcoma (FUS) and Heterogeneous Nuclear Ribonucleoprotein A2/B1 (hnRNP A2/B1) form liquid droplets, then undergo gelation and may be converted into aggregated fibrils” This sentence needs a reference.

> Response: two references have been added to support this statement

3. “First, we followed the evolution of the droplets over time by dynamic light scattering (DLS). In the case of the pH jump, initially small droplets of about 300 nm in diameter formed, growing slowly to a maximum of 1500 nm after approximately 1.5 h (Figure 1d), with an exponential time dependence of t to the power of $1/3$, as expected for particle growth by Ostwald ripening 15.” Given the size and nature of the structures I wonder if the behavior observed is truly due to Ostwald ripening (sedimentation, wetting, etc).

> Response: this is a reasonable point (also raised by another Reviewer), as the exponent of the fit depends on many factors, and would only be $1/3$ if Ostwald ripening occurred and we had a signal that is strictly proportional to the number of droplets. We note, however, that in our case there is quite some variation in the determined value (from 0.2 to 0.5), which we attribute to factors: (i) no clarity of final state, (ii) aggregation, (iii) sedimentation, and (iv) wetting. In accord, we have toned down our claim that we always encounter Ostwald ripening, and highlight now in the text that many factors contribute to the signal observed (page 5, middle).

4. How do the authors explain the following controversy: “Upon MBP cleavage, hardly any droplets are seen early on but appear much later in a practically salt-independent manner”. However, in their comparative analysis with the other constructs the formation of the structures is strongly salt-dependent. Would this not suggest that the properties are different and/or that the mechanism of structure formation is different?

> Response: this is in fact the case, the apparent lack of salt effect in the TEV-cleavage system points to a difference in LLPS mechanism, most probably to the rate-limiting nature of the cleavage reaction itself. We now point this clearly out where we make the observation (page 3, middle).

5. “These results underline that the pH jump enables instantaneous induction of LLPS under near-native conditions, whereas other approaches may support complex and artificial kinetic schemes. Therefore, we sought to generalize this approach by demonstrating its rational extension to other proteins.”

> Response: there was now question associated with these sentences

6. The authors may want to emphasize that even for pH jump experiments the used constructs was purified under (mild) denaturing conditions using 3 M urea.

> Response: whereas it is true, the protein eventually came to a pH=11.0 buffer, without urea, so we felt it is sufficient to mention this fact of purification in Methods

7. Page 8: hnRNPA2 LCD - The title of this section is cryptic and should be replaced with something more appropriate

> Response: the title has been extended to be more descriptive

8. hnRNPA2 LC-MBP: The purification protocol indicates purification via a His-Tag. Is the HisTag present in the experiments?

> Response: yes and no, the His-tag is attached to the MBP part of the constructs and thus remains on MBP and not on hnRNPA2 LCD after TEV cleavage. A half-sentence clarifying on this point has been added to Methods (section "Initiating phase separation") and the legend of Figure 1.

9. The authors may want to state the labeling efficiency for fluorescently labeled proteins in the methods section and at which ratios the proteins were used.

> Response: we have determined labeling efficiency (typically around 0.5), and now mention it in Methods (section "Fluorescent labelling of proteins").

10. The information about the 96 well plates is missing. This information would be useful.

> Response: a section has been added to Methods (section "Turbidity measurements").

11. The authors may want to consider that the problem of using absorbance measurements at OD 600 particles results in a detection limit for significant but small particle sizes.

> Response: at the outset, we screened six different wavelengths ranging from 340 nm to 600 nm. Whereas we did not observe large differences, we found it useful to mention it in the text and insert a new figure (new Figure S2) to give some guidance to the reader.

12. The method section about microscopy is incomplete and should e.g. list the objectives used with magnification, numerical aperture, etc.

> *Response: “Fluorescent and brightfield microscopy” section in Methods has been updated.*

13. I think the authors should include unrelated polypeptides, such as BSA or alpha Lactalbumin in the pH-jump approach and compare the results to the ones obtained for disordered polypeptides.

> *Response: we have carried out this control experiment (actually, with BSA and lysozyme), and now present it in a new figure (Figure S11) Further, we refer to this observation now to support our claim that the pH-jump method is the most “native like”, because these proteins, which do not function by - but under extreme conditions, undergo - LLPS, show no sign of phase separation by our novel approach (page 5, bottom - page 6, top).*

14. For Nup98 the microscopy rather suggests that the protein aggregates rather than forms liquid-like structures. How do the authors envision the growth observed by DLS? Why is the image at 2 hours so dim?

> *Response: we have re-analyzed all NUP98 microscopic images and could replace Figure 2h with a new image showing small spherical droplets (new Figure 2b).*

15. For Erd14 the authors should include controls with and without polyU, as well as polyU only. The authors should also mention more accurately the conditions, e.g. that this was in the presence of polyU.

> *Response: we have carried out these controls, and now incorporate them in a new figure (Figure S7). Further, it is now explicitly mentioned in the text that LLPS of ERD14 is observed in the presence of poly(U) and PEG (page 5, middle, and legend of Figure 2).*

16. The authors should test for the ability to dissociate the structures to test for reversibility.

> *Response: we have done this control experiment with hnRNPA2 LCD, demonstrating reversibility of phase separation (upon going from pH = 11.0 to 7.5 and then back to 11.0). This is now mentioned in the text (page 3, middle) and is shown in the new Figure S1.*

Reviewer #3

The communication from Van Lindt, et. al. shows that extreme pH values can be used to solubilize various low complexity domain proteins which have been demonstrated to undergo liquid-liquid phase separation (LLPS) under biochemical conditions which mimic native solution conditions. They have used this to jump proteins into physiological pH and demonstrated that the kinetics of phase separation is dramatically different between three major methods for inducing phase separation: jump in pH, sudden drop in denaturant (Urea, Guanidinium chloride), or cleavage of a solubilizing tag. The authors argue that the kinetics of phase separation as shown by a pH induced jump is the most “native-like” path toward phase separation given the milder conditions provided by a change in pH.

The authors report on an interesting use of a phenomenon that was demonstrated earlier in the literature, albeit for a different purpose, and this work should encourage the use of pH jumps to study phase separation. The experiments support the central argument of different kinetic mechanisms of phase separation induced by different means of inducing phase separation. This reviewer cautions against drawing a quantitative value from the DLS scattering curves. The power law fit of the curves isn't consistent all the time (although this might be due to other competing mechanisms such as aggregation) and detracts from the main point that a significantly different particle distribution exists (also showing some representative curves in the supplement would help hammer this point home as well).

> Response: thank you for this point, and we try to respond in two ways. First, we added a note to the text (as also raised by another Reviewer) that the exponent of DLS fit depends on many factors, and would only be 1/3 if Ostwald ripening occurred and we had a signal that is proportional to the number of droplets. In our system, there is quite some variation in the determined value (from 0.2 to 0.5), which we attribute to a variety of factors: (i) no clarity of final state, (ii) aggregation, (iii) sedimentation, and (iv) wetting. In accord, we have toned down our claim that we always encounter Ostwald ripening, and mention in the text these other factors contributing to the signal observed (page 5, middle).

Second, we have incorporated a new figure (Figure S8), which shows original fits of primary DLS curves, demonstrating different particle distributions under the three conditions (pH jump, urea dilution and TEV cleavage) - this is now also mentioned in the text (page 5, middle).

Secondly, this reviewer believes emphasis on a jump in pH being a “near-native” like mechanism compared to Urea should be curtailed. Such a jump from either pH 11 or 3.5 to neutral may appear to behave more “native-like” for the proteins tested here, there's no such guarantee this holds for other IDRs which undergo LLPS, and certainly won't for proteins which also contain folded domains. Rather an emphasis on how pH provides an orthogonal mechanism to salt or denaturant to study the molecular mechanism of LLPS would be much more appealing for specialists in the biophysics of LLPS as well the broader audience interested in membraneless organelles.

> Response: we appreciate this point, but would like to argue that all three methods can be considered equally close to “native like” conditions. Whereas we agree that the initial conditions (e.g. pH=11.0) are not native-like, the technique enables to jump instantaneously and with much less interference to any select condition than with the other two approaches

shown. We agree, nevertheless, that with folded proteins the initial extreme pH may cause complications (which might be more severe with 8M urea), thus we have carried out a novel control experiment with two folded proteins (BSA and lysozyme, new Figure S11). Interestingly, although these globular proteins phase separate under extreme conditions (referred to in two new references), they do not phase separate by our pH-jump method, which does reflect their physiological preferences. We have added this argument to the concluding section of the manuscript (page 5, bottom, page 6, top).

REVIEWERS' COMMENTS:

Reviewer #1 (Remarks to the Author):

I appreciate the authors' effort in the revision. The authors addressed all my comments and the manuscript has improved. I have a few minor points:

- I commented earlier on the gel band-cut outs above the now Figure 1d. I suggest moving them to the bottom of the graph, and I again urge the authors to clarify if these samples are taken at corresponding timepoints on the x-axis.
 - The scalebars in figures 1a-c are tiny and hard to read.
 - I appreciate that the authors did the extra control of seeing the effect of TEV on LLPS. I'm assuming in the rebuttal response, they mean to refer to Figure S3 and not Figure S1.
- These minor points aside, I recommend publication.

Reviewer #2 (Remarks to the Author):

The authors have addressed and answered many of my questions and I feel that the clarity of the manuscript has improved. I also appreciate the new findings, the approach, and the incorporation of rapid-mixing experiments.

However, the problem remains that the findings presented are not generally new, and others have already shown that aggregation can be inhibited by incubation at either acidic or basic pH and that a jump to more neutral pH induces assembly, or aggregation (e.g. work by Dobson C.). Another major concern that light scattering measurements give arbitrary units and not units of molarity has also not sufficiently been addressed. I agree with the authors that an apparent time trace of a complex reaction can be recorded and that the change in size may indicate growth. However, the issue of particles changing size and thus interacting non-linearly and differently with light and large particles settling during the time course of the experiment remains a concern. I would therefore urge the authors to include a cautionary note regarding the interpretation of these experiments. Another remaining concern is that the study does not actually advance our mechanistic understanding of the process. However, I am willing to support publication when the shortcomings are discussed clearly in the paper so that the reader is made aware of them.

Detailed point-by-point rebuttal to reviewer comments on our manuscript:

A generic approach to study the kinetics of liquid-liquid phase separation under near-native conditions

Reviewer #1:

I appreciate the authors' effort in the revision. The authors addressed all my comments and the manuscript has improved. I have a few minor points:

- I commented earlier on the gel band-cut outs above the now Figure 1d. I suggest moving them to the bottom of the graph, and I again urge the authors to clarify if these samples are taking at corresponding timepoints on the x-axis.

> response: gel band-cutouts are now framed, simplified (marked by "S" for supernatant and "P" for pellet) and are not incorporated below the panel by marks for their time. Figure legend has been adapted to comply with these changes.

- The scalebars in figures 1a-c are tiny and hard to read.

> response: scalebars have been corrected

- I appreciate that the authors did the extra control of seeing the effect of TEV on LLPS. I'm assuming in the rebuttal response, they mean to refer to Figure S3 and not Figure S1.

> response: this is correct, in the manuscript we correctly referred to Figure S3.

Reviewer #2:

The authors have addressed and answered many of my questions and I feel that the clarity of the manuscript has improved. I also appreciate the new findings, the approach, and the incorporation of rapid-mixing experiments.

However, the problem remains that the findings presented are not generally new, and others have already shown that aggregation can be inhibited by incubation at either acidic or basic pH and that a jump to more neutral pH induces assembly, or aggregation (e.g. work by Dobson C.).

> response: we have added a note to the first section of Results that pH jump was already used for studying LLPS (ref. 15) and aggregation, as demonstrated by C. Dobson (new ref 16). Further, we also refer now a new paper by the Julicher group (Adame-Arana 2020 Biophys. J., ref. 21) that provides a theoretical framework for the general pH dependence of LLPS. As a caveat to this note, however, we have to be aware that in the aggregation experiments carried out by Dobson and colleagues, the purpose of applying extreme pH was not to keep a protein in solution - away from aggregating - and then initiate aggregation by returning to native state, rather to unfold the protein and make it more aggregation prone. In this sense, their motivation of applying the pH jump is rather the opposite to what we suggest in our manuscript.

Another major concern that light scattering measurements give arbitrary units and not units of molarity has also not sufficiently been addressed. I agree with the authors that an apparent time trace of a complex reaction can be recorded and that the change in size may indicate growth. However, the issue of particles changing size and thus interacting non-linearly and differently with light and large particles settling during the time course of the experiment remains a concern. I would therefore urge the authors to include a cautionary note regarding the interpretation of these experiments.

> response: a detailed note has been added at the middle of Results section, when we conclude turbidity (OD600) experiments and suggest that further insight can be gained from carrying out DLS measurements.

Another remaining concern is that the study does not actually advance our mechanistic understanding of the process.

> response: we suggest now at the end of Discussion that recording the kinetic traces as suggested in the manuscript have to be extended by further studies, in order to gain mechanistic understanding of LLPS kinetics in atomic detail.